# *In Vitro* and *In Vivo* Inhibition of MATE1 by Tyrosine Kinase Inhibitors

**DOI:** 10.3390/pharmaceutics13122004

**Published:** 2021-11-25

**Authors:** Muhammad Erfan Uddin, Zahra Talebi, Sijie Chen, Yan Jin, Alice A. Gibson, Anne M. Noonan, Xiaolin Cheng, Shuiying Hu, Alex Sparreboom

**Affiliations:** 1Division of Pharmaceutics and Pharmacology, College of Pharmacy, The Ohio State University, Columbus, OH 43210, USA; uddin.33@osu.edu (M.E.U.); talebi.9@osu.edu (Z.T.); jin.1134@osu.edu (Y.J.); gibson.972@osu.edu (A.A.G.); 2Division of Medicinal Chemistry and Pharmacognosy, College of Pharmacy, The Ohio State University, Columbus, OH 43210, USA; chen.3428@buckeyemail.osu.edu (S.C.); cheng.1302@osu.edu (X.C.); 3Department of Internal Medicine, Division of Medical Oncology, College of Medicine, The Ohio State University, Columbus, OH 43210, USA; Anne.Noonan@osumc.edu; 4Division of Outcomes and Translational Sciences, College of Pharmacy, The Ohio State University, Columbus, OH 43210, USA; hu.1333@osu.edu

**Keywords:** MATE1, TKIs, knockout mice, renal elimination, drug-drug interactions

## Abstract

The membrane transport of many cationic prescription drugs depends on facilitated transport by organic cation transporters of which several members, including OCT2 (*SLC22A2*), are sensitive to inhibition by select tyrosine kinase inhibitors (TKIs). We hypothesized that TKIs may differentially interact with the renal transporter MATE1 (*SLC47A1*) and influence the elimination and toxicity of the MATE1 substrate oxaliplatin. Interactions with FDA-approved TKIs were evaluated in transfected HEK293 cells, and *in vivo* pharmacokinetic studies were performed in wild-type, MATE1-deficient, and OCT2/MATE1-deficient mice. Of 57 TKIs evaluated, 37 potently inhibited MATE1 function by >80% through a non-competitive, reversible, substrate-independent mechanism. The urinary excretion of oxaliplatin was reduced by about 2-fold in mice with a deficiency of MATE1 or both OCT2 and MATE1 (*p* < 0.05), without impacting markers of acute renal injury. In addition, genetic or pharmacological inhibition of MATE1 did not significantly alter plasma levels of oxaliplatin, suggesting that MATE1 inhibitors are unlikely to influence the safety or drug-drug interaction liability of oxaliplatin-based chemotherapy.

## 1. Introduction

Membrane-transporters are key regulators of selective cellular permeability [1], and mediate the passage of many endogenous metabolites such as amino acids and nucleosides as well as small-molecule xenobiotics across the plasma membrane [2]. Therefore, they are key determinants in normal physiology and pathophysiology, as well as therapeutic response to drugs. At the cellular level, transporter-mediated uptake or efflux can lead to emergence of drug sensitive or resistant phenotypes in target cells [3], and as such affect therapeutic efficacy. On the other hand, transporter-mediated uptake in non-target tissues can result in drug-related toxicities [4], and several of the transporters involved in this processes have emerged as critical determinants of drug absorption, disposition, therapeutic efficacy, adverse drug reactions, and drug-drug interactions.

About 40% of approved prescription drugs are positively charged at neutral pH (“organic cations”), and the membrane transport of such agents is dependent on facilitated uptake carriers. The organic cation transporters OCT2 (*SLC22A2*) and MATE1 (*SLC47A1*) have particular relevance in this connection, since they are highly expressed at the basolateral and luminal membranes of renal tubular cells, respectively, and these proteins are considered major transporters in the secretion of organic cations from the circulation into the kidney and then from the kidney into the tubular lumen. The tissue distributions of OCT2 and MATE1 in mice are generally consistent with that in humans, with the exception of MATE2-K being absent in the kidney of mice [5]. Human and mouse MATE1 exhibit mutual sequence identity (78%) with similar characteristics [6]. Substrates for MATE1 are typical organic cations, e.g.,tetraethylammonium (TEA), 1-methyl-4-phenylpyridinium (MPP), metformin, cimet-idine, procainamide, and oxaliplatin, which are also found to be transported by mouse MATE1 [7]. Defects in OCT2 or MATE1 function resulting from reduced-function alleles in the *SLC22A2* and *SLC47A1* genes can lead to impaired drug elimination and cause increases in circulating concentrations of xenobiotic substrates [8]. These prior genetic studies suggest that unintentional alteration of organic cation transporter function, for example by the use of drugs with OCT2- and/or MATE1-inhibitory properties, can potentially lead to deleterious phenotypic changes in patients [9].

Previous studies have identified a number of widely used anticancer drugs in the class of tyrosine kinase inhibitors (TKIs) as particularly potent inhibitors of OCT2 [10] as well as the phylogenetically-linked transporters OCT1 [11] and OCT3 [12]. However, a systematic approach to evaluate the ability of TKIs to interact with MATE1 and subsequently affect endogenous homeostasis and xenobiotic handling of substrate drugs such as oxaliplatin is still lacking. In the current study, we characterized the interaction of FDA-approved TKIs with MATE1 *in vitro* in cells, *in silico* by molecular docking simulations, and *in vivo* in genetically-engineered mouse models and a patient with cancer.

## 2. Materials and Methods

### 2.1. Cellular Models and Cell Culture Conditions

Parental human embryonic kidney (HEK293) cells were obtained from American Type Culture Collection (ATCC; Manassas, VA, USA). The cDNAs for the mouse and human plasmids of OCT2 or MATE1 were obtained from Origene (Rockville, MD, USA), and the reconstructed cDNAs were subcloned into an empty vector containing pcDNA5/FRT. The vector was transfected into HEK293 cells using the Flp-In system (Invitrogen, Waltham, MA, USA) and selected for expression using geneticin (G418). Cells were cultured in DMEM supplemented with 10% FBS and grown in a humidified incubator containing 5% CO_2_ at 37 °C. Lipofectamine 2000 or LTX (Life Technologies, Rockville, MD, USA) was used for transient transfections. TKIs were obtained from Sigma-Aldrich (St. Louis, MO, USA) or Selleckchem (Houston, TX, USA). Radiolabeled compounds were obtained from American Radiolabeled Chemicals (St. Louis, MO, USA) or Moravek (Brea, CA, USA). Oxaliplatin was obtained from Tocris (Minneapolis, MN, USA), and cisplatin from Sigma-Aldrich (St. Louis, MO, USA).

### 2.2. Uptake Assays

Uptake experiments in HEK293 cells overexpressing mouse (m) and human (h) OCT2 or MATE1 were performed with radiolabeled tetraethylammonium (TEA) as the model substrate, using standard methods [13,14] in the presence or absence of TKIs. The cell culture and uptake conditions for cells expressing mMATE1 or hMATE1 were described previously [12]. Since several xenobiotics can inhibit OCT2- and/or MATE1-mediated transport of certain substrates drugs with higher affinities than TEA [9], the ability of select TKIs to inhibit the uptake of metformin and oxaliplatin was also evaluated. To ensure that inhibition via high-affinity binding sites can be detected [9], substrate concentrations were selected that are ≥100-fold below the K_m_ values for OCT2 and MATE1. Considerations of inhibitor concentration selection in the *in vitro* studies were based on prior criteria [9] in order to exceed the anticipated unbound plasma concentrations of investigated TKIs and demonstrate inhibitory potential. Since the inhibitory potential of various xenobiotics against OCT2 and MATE1 is dependent on preincubation with perpetrator drugs [10,15,16], a 15-min preincubation period with the TKIs was employed unless stated otherwise. All results were normalized to uptake values in cells transfected with an empty vector or DMSO-treated groups.

In the uptake experiments, cells were seeded in 12- or 24-well plates (6-well plates for oxaliplatin) in phenol red-free DMEM containing 10% FBS, and were incubated at 37 °C for 24 h. After removal of the culture medium and rinsing with PBS, cells were preincubated with either DMSO or inhibitors followed by addition of substrate and uptake measurement after 10–15 min for TEA (2 µM) or metformin (5 µM) and 60 min for oxaliplatin (50–75 µM). MATE1-overexpressed cells were pre-incubated with transport media containing 30 mM ammonium chloride for 20 min at 37 °C following previously published protocol (pH 7.4 for hMATE1 [17,18,19] and pH 8.4 for mMATE1 [20,21]) to ensure interactions can be evaluated between outward-facing MATE1 and substrate uptake before adding inhibitors. Then, preincubation media was removed and cells were incubated with transport media (NH_4_Cl free) containing radiolabeled compounds. The composition of transport buffer was as follows: 145 mM NaCl, 3 mM KCl, 1 mM CaCl_2_, 0.5 mM MgCl_2_, 5 mM D-Glucose, and 5 mM HEPES. Although the physiological function of MATE1 is to support luminal efflux, previous studies have shown that kinetic parameters of MATE1-mediated transport are not significantly different between the outward (uptake) and inward (efflux) orientations [22]. The uptake of TEA and metformin was measured by liquid scintillation counting, as described previously [23]. For uptake studies involving platinum drugs, cells were collected with TrypLE, and the pellets were lysed overnight in 0.2% nitric acid. After sonication, total platinum levels were determined by flameless atomic absorption spectrometry [23].

### 2.3. Gene Expression Analysis

RNA was extracted using the Qiagen kit from treated cells in 12-well plates or kidney tissues obtained from untreated or treated mice. Taqman primers Hs00217320_m1 and Mm00840361_m1 were obtained from Thermo Fisher Scientific (Waltham, MA, USA) and used for the hMATE1 and mMATE1 genes. The primers Hs02786624_g1 and Mm99999915_g1 were used as the house-keeping control genes for hGAPDH and mGAPDH, respectively.

### 2.4. Computational Modeling

Ligand-based pharmacophore modelling and molecular docking were performed with Schrodinger Suite 2018 [24,25]. TKIs were classified as active and inactive based on their cellular uptake activities, where TKI-mediated inhibition of MATE1 function was considered ‘active’ if transport function was reduced to ≤10%of baseline values, and ‘inactive’ was defined as inhibition resulting in residual cellular uptake values of >50% of baseline values. A total 43 TKIs was used in pharmacophore construction. Common chemical features were extracted through the alignment of all active TKIs to capture the essential interactions between the ligands and a potential protein target. Molecular docking predicts the preferred binding conformation and orientation of a ligand within a protein target and the binding poses are evaluated based on an energy function also known as scoring function. Machine learning-based QSAR (ML-QSAR) models were constructed as binary classification with support vector machine (SVM) and artificial neural network (ANN) [26,27]. A group of 58 tested compounds was randomly divided into a training and a testing dataset in an 8:2 ratio. All TKIs were converted to Morgan circular fingerprints. For the ANN-based QSAR models, 20 runs were taken, and the ANN was trained with 150 epochs in each run. Linear discriminative analysis (LDA) and t-distributed stochastic neighbor embedding (t-SNE) were performed to explore the chemical space spanned by these TKIs. In the LDA analysis, we randomly selected 100 known allosteric modulators or competitive inhibitors, and then converted their SMILES representations to Morgan Circular fingerprint using RDKit. The 2D chemical structures of the tested TKIs were also converted to Morgan Circular fingerprint. The molecular fingerprints and labels of all compounds were fitted with the LDA model using Scikit-learn (sklearn), and results were plotted as probability density distributions. In the t-SNE analysis, principal component analysis (PCA) was performed for the first 20 dimensions of the molecular fingerprints for all compounds using sklearn. The PCA results were then fed into t-SNE model using sklearn, and the result was visualized with a scatter plot. The source codes for ML-QSAR, similarity comparison, LDA and t-SNE are available at https://github.com/sijiechenchenchen (Access date: 24 September 2021).

### 2.5. Murine Pharmacokinetic Studies

For pharmacokinetic studies, plasma and tissue samples were collected from male wild-type mice (8–12 weeks old), and age-matched MATE1-deficient [MATE1(−/−)] mice, and mice additionally deficient for OCT1 and OCT2 [OCT1/2/MATE1(−/−)], following an established protocol [28]. All animals were backcrossed on an FVB strain, were given a standard diet and water *ad libitum*, and housed and handled in accordance with the University Laboratory Animal Resources (ULAR) Animal Care and Use Committee at The Ohio State University (2015A00000101-R1). The expected gene deletions in OCT1/2/MATE1(−/−) mice were verified by PCR [11], and these animals were observed to be viable and fertile without detectable serum biochemical abnormalities (Appendix A). Oxaliplatin was administered to mice as a single i.p. dose (10 mg/kg), with oxaliplatin dissolved in sterile PBS containing 5% glucose. Dasatinib (15 mg/kg) was formulated in 80 mM citric acid (pH 3.1) and given orally 30 min prior to the administration of oxaliplatin. Concentrations of total platinum in plasma and tissues were measured by flameless atomic absorption spectrometry, as previously reported [14,29], by interpolation of unknown readings on linear calibration curves prepared in drug-free nitric acid (0.2%) using linear-least squares regression analysis.

In order to examine the influence of transporter deficiency on the elimination of total platinum after oxaliplatin administration, male wild-type, MATE1(−/−), and OCT1/2/MATE1(−/−) mice were kept individually in Nalgene metabolic cages for a period of 3 days prior to oxaliplatin administration (10 mg/kg, i.p.). Animals had free access to a standard diet and water, and were housed in a temperature- and light-controlled environment. Changes in the animals’ appearance (e.g., kyphosis and altered grooming), behavior (e.g., altered nesting), and/or activity (e.g., altered exploring) were monitored throughout the experiments. The bodyweight of each individual mouse was recorded before and during the experiment. Urine samples were collected in sterile 1.5 mL Eppendorf tubes at 8, 24, 48, and 72 h post-administration of oxaliplatin. Drug levels in urine were analyzed by a validated method based on flameless atomic absorption spectrometry [14], and urinary excretion of oxaliplatin was expressed as a percentage of the total administered dose.

### 2.6. Immunoblotting Assays

Kidneys were isolated from mice, then minced and incubated in 500 µL of RIPA buffer (Cell Signaling Technology, Danvers, MA, USA) containing 1% SDS and protease and phosphatase inhibitors. After sonication, the lysate was incubated for 10 min on ice, and then centrifuged at 15,000× *g* for 15 min. The supernatant was re-centrifuged and used for determination of total protein content by a BCA assay kit. For the immunoblotting, the membrane was blocked with 5% skim milk in 0.05% PBS-tween, and then incubated overnight with a primary antibody (NGAL # 0351) obtained from Santa Cruz Biotechnology (Dallas, TX, USA), and with an HRP-conjugated secondary anti-rabbit antibody (7074) obtained from Cell Signaling Technology (Danvers, MA, USA).

### 2.7. Human Pharmacokinetic Studies

The study subject was a patient with metastatic colorectal cancer receiving a once daily oral dose of dasatinib (100 mg) before initiation of mFOLFOX6-based therapy, which included oxaliplatin (85 mg/m^2^), leucovorin (400 mg/m^2^), and 5-fluorouracil (400 mg/m^2^ by i.v. bolus and 2400 mg/m^2^ by infusion over 46 h). Dasatinib was administered in combination with standard doses of oxaliplatin given every 14 days, 24 h before and 30 min before each oxaliplatin infusion. In order to assess the degree of MATE1 inhibition by dasatinib, analyses of N-methylnicotinamide (see below) were performed in plasma samples collected at serial time points (up to 24 h after the end of infusion). Plasma samples were also subjected to dasatinib analysis by liquid chromatography with tandem mass spectrometric detection (LC-MS/MS) [30] and analyzed for total platinum levels by flameless atomic absorption spectrometry [31]. The study protocol (ClinicalTrial.gov Identifier: NCT04164069) was approved by the institutional review board of the -The Ohio State University Comprehensive Cancer Center (OSU IRB #2019C0141, approval date 01 March 2019), and the patient provided written informed consent prior to enrollment.

### 2.8. Quantification of N-Methylnicotinamide

Levels of N-methylnicotinamide (NMN; purity >98%, Sigma Aldrich, St. Louis, MO, USA) were determined using LC-MS/MS using a Vanquish UHPLC coupled with a Quantiva triple quadrupole mass spectrometer (Thermo Fisher Scientific). An Accucore aQ C18 column (150 × 2.1 mm, dp = 2.6 μm) with a C18 AQUASIL guard cartridge (2.1 mm × 10 mm, dp = 3 μm) were employed for separation of the analytes of interest. The column and autosampler were maintained at temperatures of 40 °C and 4 °C, respectively. The mobile phase was composed of solvent A (0.1% formic acid in distilled water) and solvent B (0.1% formic acid in acetonitrile: methanol, 50:50, *v*/*v*), and a gradient elution was used for 5.0 min at a flow rate of 0.4 mL/min. The gradient conditions were as follows: 0–0.5 min, 0% B; 0.5–1.5 min, 0% to 20% B; 1.5–2.3 min, 20% B; 2.3–3.8 min, 20–95% B; 3.8–4.2 min, 95% B; 4.2–5.0 min, 0% B. Aliquots of 5 μL of the extracted samples were injected. The following parameters were set for the mass spectrometer: sheath gas, 40 Arb; aux gas, 12 Arb; sweep gas, 3.3 Arb; ion transfer tube temperature, 350 °C, and vaporizer temperature, 375 °C. The ion source was operated using heated ESI with the ion spray voltage set at 3500 V in positive ion mode. The collision gas argon was used at a pressure of 1.5 mTorr. An optimized selective reaction monitoring (SRM) mode was applied for the quantitation with the following parameters: *m*/*z* 137.062 > 94.012, collision energy at 20.05 V for N-methylnicotinamide and *m*/*z* 140.050 > 97.082, collision energy at 21.68 V for the internal standard, 3-carbamoyl-1-methyl-d3-pyridinium chloride (purity: >98%, Toronto Research Chemicals, North York, ON, Canada). A protein-precipitation method was used to extract N-methylnicotinamide from plasma samples. In brief, prior to analysis, frozen samples were thawed at room temperature, plasma aliquots of 10 μL were transferred to a 0.5-mL Eppendorf tube followed by the addition of 20 μL of an internal standard working solution (100 ng/mL) and 70 μL of methanol. The samples were vortex-mixed and centrifuged at 13,000 rpm for 9.5 min at 4 °C. Next, 60-μL aliquots of the organic layer were transferred into WebSeal 96-well plates covered with a Webseal mat (Thermo Fisher Scientific), and a 5-μL volume of each was injected into the LC-MS/MS system. The lower limit of quantitation for N-methylnicotinamide was determined to be 1 ng/mL, and linear calibration curves ranged from 1 to 1000 ng/mL. The within-run and between-run precisions were within 5.27%, while the accuracy ranged from 93.5 to 104%.

### 2.9. Statistical Analyses

All data are presented as mean ± standard error of the mean (SEM), and experimental results from uptake studies were normalized to total protein content and baseline values, and expressed as a percentage. All experiments were performed using multiple replicates and were performed independently on at least two independent occasions. An unpaired two-sided Student’s t-test with Welch’s correction was used for comparisons between two groups (control/baseline vs. treatment/genotype), and a one-way ANOVA with Dunnett’s post-hoc test was used for comparing more than 2 groups. *p* < 0.05 was used as the statistical cut-off across all analyses.

## 3. Results and Discussion

### 3.1. TKI-Mediated Inhibition of MATE1 In Vitro

In our previously reported screens of inhibitors against the organic cation transporters OCT1, OCT2, and OCT3, several TKIs were identified as agents with potent transporter-modulatory properties [10,11,12]. Due to their unique inhibitory activity against these transporters, we performed a screen of FDA-approved TKIs and sought to determine whether these agents are also modifiers of transport function in cells overexpressing the related transporter, MATE1. In order to determine the extent of interactions between TKIs and MATE1, the uptake of the prototypical substrate TEA was evaluated in cells overexpressing hMATE1 in the presence or absence of a pre-incubation with each individual TKI. The known MATE1 inhibitor cimetidine was used as a positive control inhibitor. This screen revealed that all 57 TKIs, except for alectinib, inhibited MATE1 function by at least 25% with an inhibitor to substrate concentration ratio of 5:1, and more than half of the TKIs inhibited by more than 80% (Figure 1A). The number of TKIs with MATE1-inhibitory properties and the observed extent of inhibition was substantially larger than what was predicted based on publicly available data (Appendix A).

Subsequent investigation with various anti-leukemic TKIs, including dasatinib, ibrutinib, imatinib, nilotinib, and/or ponatinib, indicated that the TKI-MATE1 interaction is dependent on the pre-incubation time (Figure 1B), and is only partially reversible such that MATE1 function is not fully restored even after an 8-h washout period (Figure 1C). This suggests that even short-term exposure to TKIs followed by their removal can result in sustained inhibition of MATE1 for prolonged time periods. Since certain inhibitors of organic cation transporters function in a substrate-dependent manner [32], we verified that several TKIs retained MATE1-inhibitory potential when using metformin as an alternative, secondary substrate (Figure 1D). Furthermore, a Dixon plot of the reciprocal velocity against the TKI concentration to derive inhibition constants (Ki) indicated that the mechanism of inhibition is non-competitive (Figure 1E). Conclusion is consistent with a previous observation that certain anti-leukemic TKIs, including nilotinib (Ki, 1.60 µM), can exert selectively potent inhibitory effects against MATE1 at clinically-relevant concentrations [15].

### 3.2. Computational Evaluation of MATE1-TKIs Interaction

To gain insight into the mechanistic basis of the finding that an unexpectedly large number of TKIs have potent MATE1-inhibitory properties, two ligand-based models were constructed, including pharmacophore models and ML-QSAR models (Figure 2A). The two ML-QSAR models using artificial neural network (ANN) and support vector machine (SVM) yielded q^2^ values of 0.83 and 0.75 in the test dataset, respectively (Figure 2B). The good accuracy of the ML-QSAR models indicates that the inhibitory activities of TKIs are correlated with their chemical structural features, which in turn suggests a common inhibitory mechanism. To probe the mechanisms by which these TKIs interact with human MATE1, we generated a pharmacophore model with the maximum hypothesis match of the 24 ‘active’ TKIs that reduced the function of MATE1 by >90%. The low accuracy of the model (q^2^ = 0.51) indicates, however, that these TKIs do not share a common pharmacophore and, hence, are unlikely bind to the same well-defined pocket (site) on MATE1. The main reason for the poor performance of this pharmacophore model is related to the diverse structures of the active TKIs, as indicated by the low Tanimoto similarity scores (Figure 2C), where most TKIs show less than 0.3 similarity across the three comparisons.

The notion that TKIs are non-competitive inhibitors of MATE1 suggests that the TKIs do not occupy the orthosteric binding site for substrates. To investigate if the active TKIs act as allosteric modulators of MATE1, we compared their chemical characteristics with those of existing allosteric drugs. We randomly selected 100 compounds belonging to allosteric modulators or competitive inhibitors, and conducted linear discriminant analysis (LDA) to classify these compounds (Figure 2D) [33]. Our results show that LDA can distinctly separate the allosteric modulators from the competitive inhibitors, and our tested active TKIs populate the purple distribution region representing allosteric modulators. In addition, we employed t-distributed Stochastic Neighbor Embedding (t-SNE) to visualize if the TKIs, allosteric modulators, and approved drugs reside in the same chemical space (Figure 2E). Both LDA and t-SNE show that the active TKIs and allosteric drugs are clustered together, indicating that the active TKIs resemble allosteric modulators in terms of their physicochemical properties.

To further probe if and how these TKIs interact with MATE1, we computationally docked them to an atomic structural model of MATE1 predicted by AlphaFold2 (access code AF-Q96FL8-F1) [34]. The docking result does not show a consensus binding site/pose for these TKIs, consistent with the above pharmacophore model. Additionally, the docking scores show no significant difference among the active, neutral and inactive TKIs. The best binding poses of the active, inactive, and neutral TKIs show that their binding space overlaps, all occupying a large vacant volume between the C- and N-lobes of MATE1 (Figure 2F). Overall, these calculations suggest that the mechanism by which TKIs inhibit MATE1 is unlikely associated with binding to a common and well-defined binding pocket on the MATE1 protein.

### 3.3. Influence of TKIs on MATE1 Regulation In Vitro

Next, we evaluated the influence of MATE1-inhibitory TKIs on post-translational and transcriptional regulatory pathways as potential explanations for the observed inhibitory profiles. We previously reported that the transporter OCT2 is sensitive to inhibition by several FDA-approved TKIs through a mechanism that involves YES1-mediated tyrosine phosphorylation [11] Since OCT2 shares various structural features and has overlapping substrate specificity compared with MATE1 [35], we hypothesized that the activity of MATE1 might also be dependent on kinase-mediated tyrosine phosphorylation. However, a recent unbiased MS-based phospho-proteomics screen did not identify MATE1 as a tyrosine-phosphorylated protein [10]. The absence of a common inhibitory mechanism by which TKIs can modulate the function of OCT2 and MATE1 is also consistent with the notion that the inhibitory properties of a subset of TKIs against these 2 transporters are not correlated (Figure 3A). Indeed, compared to OCT2, a highly distinct TKI-mediated inhibitory profile was observed for MATE1, with some TKIs (e.g., bosutinib, sunitinib) potently inhibiting both transporters and some (e.g., regorafenib, sorafenib) having no influence on OCT2 function. These findings are consistent with the observation that silencing of YES1 by siRNA in HEK293 cells influenced the function of OCT2 but not MATE1 (Figure 3B), and with the notion that MATE1 lacks the proline-rich (PXXPR) sequence present in OCT2 that binds the Src Homology 3 domain present in YES1 [10] (Figure 3C).

An alternative route by which TKIs affect MATE1 function is through effects on transcription factors that result in gene expression changes. Previous studies have suggested a possible role for PPARα [36,37], and basal transcription of the MATE1 gene is regulated by FXR [38] and by binding of Sp1 close to the transcription start site [39], by binding of AP-1/AP2-rep to the promoter region [40], and by Nkx-2.5, SREBF1, and USF-1 [41]. Of these transcription factors, Sp1 is of particular interest because exposure to nilotinib and dasatinib is known to influence Sp1 expression and binding to its target genes as well as influence kinases that phosphorylate Sp1 via mechanisms unrelated to their primary targets [42]. To gain preliminary insights, we performed a transcriptomic analysis by RT-PCR of MATE1-overexpressing cells exposed to TKIs *in vitro* and found that dasatinib and nilotinib (in increasing order) downregulated the expression of MATE1 (Figure 3D). Although dasatinib completely inhibited the MATE1-mediated uptake of TEA, the suppression of gene expression observed in the presence of dasatinib was ~50%, suggesting that inhibition might occur via different mechanisms. Further investigations are required to evaluate the TKI-, dose-, and time-dependence as well as the MATE1 gene expression analysis in other experimental model such as renal proximal tubular cells or in mice treated with different TKIs, and examine the mechanistic basis of these findings.

### 3.4. Effect of TKI Treatment on MATE1 Function In Vivo

In advance of evaluating the influence of TKIs on MATE1 function in mice, the comparative concentration associated with 50% inhibition of transport (IC_50_) for human and mouse MATE1 was determined for several TKIs, including ibrutinib, imatinib, and nilotinib. These studies indicated that mouse MATE1 was more resistant to TKI-mediated inhibition than the human transporter; for example, the IC_50_ for nilotinib against human MATE1 was >100 times lower than that against mouse MATE1 (0.38 vs. 45.4 µM, respectively) (Figure 4A). It is noteworthy that the same pattern of inhibition was observed for the known, non-TKI inhibitor of MATE1, cimetidine (IC_50_, >50 µM). The finding that the MATE1-inhibitory property of certain xenobiotics is species-dependent is not unprecedented [43], and has been specifically reported previously for imatinib-mediated inhibition of dopamine transport (IC_50_ for human MATE1, 1.1 µM; IC_50_ for mouse MATE1, 101 µM) [43]. This suggests that phenotypic changes observed in mice could potentially under-predict observations made in humans.

To directly assess the influence of dasatinib on the function of MATE1 *in vivo*, levels of the endogenous dual OCT2/MATE1 substrate 1-N-methyl-nicotinamide (NMN) were measured as a potentially TKI-sensitive biomarker at baseline in mice that were either wild-type or deficient in MATE1 [44], OCT1 and OCT2 (OCT1/2), which together are the functional equivalent of human OCT2 [45], or both MATE1 and OCT1/2 [46]. We found that deficiency of either MATE1 or OCT1/2 was accompanied by significantly elevated levels of NMN in plasma (Figure 4B), and that simultaneous deficiency of MATE1 and OCT1/2, which form a functional unit with MATE1 in the kidney to regulate the tubular secretion of organic cations, resulted in further elevation of baseline levels. These findings suggest that NMN, a natural metabolite of niacin (or nicotinamide), serves as a *bona fide* biomarker for directional renal organic cation transport, a conclusion that is in line with a recent clinical report [47].

We next evaluated the impact of dasatinib on concentrations of NMN in mice and found that oral administration of the TKI at a dose of 15 mg/kg resulted in a transient, statistically significant increase in the plasma levels of NMN in wild-type mice to a degree that resembles those observed in MATE1-deficient mice at baseline (Figure 4B). Further investigation is required to examine the direct contribution of OCT2 and MATE1 to the clearance of 1-NMN, and determine whether the relatively modest changes in plasma levels of 1-NMN in MATE1-deficient mice compared to OCT2-deficient mice are associated with enhanced renal retention in the former animals. Similar observations were made in a patient with colorectal cancer who received a single oral 100-mg dose of dasatinib (Figure 4C). The average circulating concentrations of dasatinib in this patient were consistent with previously reported findings in humans receiving dasatinib-based treatment [48], were similar to those observed in our preclinical studies in mice (Figure 4D), and suggest that these levels are in the range required for potent and sustained inhibition of MATE1 as predicted from *in vitro* model systems. The plausibility of dasatinib-mediated *in vivo* inhibition of MATE1 is further underscored by the notion that TKIs tend to accumulate in cells (administration of imatinib 25 µM resulted in 4.2 mM intracellular concentrations in K562 cells) [49] and dasatinib is known to extensively accumulate in the mammalian kidney, the proposed site of interaction, with reported kidney-to-blood concentration ratios of >10 [50]. In view of the structural dissimilarity between TKIs, extending conclusions derived from studies with dasatinib to other TKIs is not necessarily appropriate, and suggests that further investigations are required to evaluate whether the mechanism of MATE1 inhibition is dependent on the TKI involved. In addition, while TKIs such as dasatinib potently inhibit MATE1 function *in vitro*, suggesting that concurrent treatment with such TKIs warrants caution, these interactions are not necessarily associated with a deleterious effect on renal function.

### 3.5. Modulation of Oxaliplatin Pharmacokinetics by TKI Treatment

To further evaluate the influence of TKIs on the function of MATE1 *in vivo*, the plasma pharmacokinetic profile and urinary excretion of the anticancer drug oxaliplatin were examined in mice. Oxaliplatin is a small-molecule platinum coordination complex used for the treatment of several gastrointestinal cancers that is cleared from plasma predominantly by renal elimination, with urinary excretion accounting for >50% of the dose and fecal excretion for about 2% of the dose [51]. Previous reports demonstrated that oxaliplatin is a transported substrate of mouse, rat, and human OCT2 [14,52] as well as MATE1 [46,53], and these transporters collectively provide a mechanistic account of the renal tubular secretion of oxaliplatin in rodents and humans (Figure 5A). Preliminary *in vitro* studies provided confirmation of the notion that oxaliplatin is a transported substrate of mouse and human OCT2 and mouse and human MATE1, where overexpression of the murine (Figure 5B) and human (Figure 5C) transporters was associated with 15-fold and 3-fold increases in uptake, respectively, compared with cells transfected with an empty vector control. Importantly, cells pre-treated with dasatinib showed a dramatically impaired ability to transport oxaliplatin.

In advance of the pharmacokinetic studies with oxaliplatin, we verified the utility of the MATE1-deficient mouse model by demonstrating its increased sensitivity to nephrotoxicity associated with the related agent cisplatin that results from an increase in residence time and extent of accumulation in proximal tubular cells associated with impaired MATE1 function. Indeed, we found that the administration of cisplatin to MATE1-deficient mice was associated with increases in several commonly used biomarkers of drug-induced acute kidney injury, including blood urea nitrogen (BUN) and neutrophil gelatinase-associated lipocalin (NGAL) compared with the values observed in wild-type mice (Appendix A). These findings are consistent with prior observations in similar genetically-engineered mouse models [54,55,56] or studies involving the administration of cisplatin together with inhibitors of MATE1 [57,58], and are in line with observations made in patients receiving treatment with cisplatin who carry impaired function variants in *SLC47A1*, the gene that encodes MATE1 [59,60,61].

The notion that oxaliplatin lacks nephrotoxic properties similar to those observed with cisplatin has been ascribed to comparative differences in transport kinetics for MATE1 in both mice and humans, and with a corresponding increase in luminal efflux into the urine [32]. This, in turn, provides a plausible explanation for the differential extent of total drug accumulation in the kidney, which is much less for oxaliplatin than for cisplatin [32,33]. Based on this prior knowledge, it has been speculated that genetic or pharmacological inhibition of MATE1 might also be associated with an oxaliplatin-mediated potentiation of nephrotoxicity [62]. In contrast, we found that while the urinary excretion of oxaliplatin was substantially decreased by MATE1-deficiency over a 3-day collection period (Figure 6A), this was not accompanied by detectable changes in treatment-related weight loss (Figure 6B), a general marker of toxicity, or by specific markers of nephrotoxicity (Figure 6C,D). Similar observations were made in animals that were additionally deficient in OCT1 and OCT2 (Figure 6A–D). The translational significance of these findings was confirmed by the demonstration that the pretreatment of mice with dasatinib had no influence on markers of oxaliplatin-induced nephrotoxicity (Appendix A). Interestingly, the plasma pharmacokinetic profile of oxaliplatin was unchanged by pretreatment with a single dose of dasatinib in wild-type mice (Figure 6E), and in the patient with colorectal cancer (Figure 6F; Appendix A). This observation was made regardless of MATE1 or OCT1/2 genotype (Appendix A), and the lack of a detectable plasma pharmacokinetic interaction was also noted for cisplatin when given in combination with nilotinib (Appendix A). The paradoxical notion that urinary excretion of oxaliplatin can be impaired as a result of dysfunctional transport by MATE1 without concurrent changes in the apparent systemic clearance is possibly related to shunting of elimination. A related phenomenon has been documented previously for several other drugs where altered liver uptake or biliary secretion due to a genetic defect is associated with increases in the extent of urinary excretion [63,64]. Although additional investigation will be required to assess the precise mechanistic basis and the pharmacodynamic implications of this phenomenon, the observations made in mice and humans provide evidence that MATE1 inhibitors are unlikely to influence the safety or drug-drug interaction liability of oxaliplatin-based chemotherapy.

## 4. Conclusions

In the present study, we identified MATE1 as a transporter that is sensitive to potent inhibition by a remarkably large number of small molecule drugs in the class of TKIs, and demonstrated through functional validation studies using genetic and pharmacological approaches that the observed inhibitory properties are potentially related to an effect on transcription. In addition, we found that some of these TKIs can inhibit renal MATE1 function *in vivo* as evidenced from their ability to modulate urinary excretion of the prototypical substrate, oxaliplatin, as well as systemic levels of the endogenous biomarker, NMN. These findings provide novel insight into the regulation of MATE1 and suggest that caution is warranted with polypharmacy regimes involving the use of certain MATE1 substrates given in combination with TKIs. Developing a comprehensive physiologically based pharmacokinetics (PBPK) model is required in order to predict TKIs and MATE1 mediated drug-drug interactions in humans. This is particularly relevant in view of the fact that a large proportion of approved prescription drugs are organic cations, and that the membrane transport of many of these agents depends on facilitated carriers such as MATE1 [65].

## Figures and Tables

**Figure 1 pharmaceutics-13-02004-f001:**
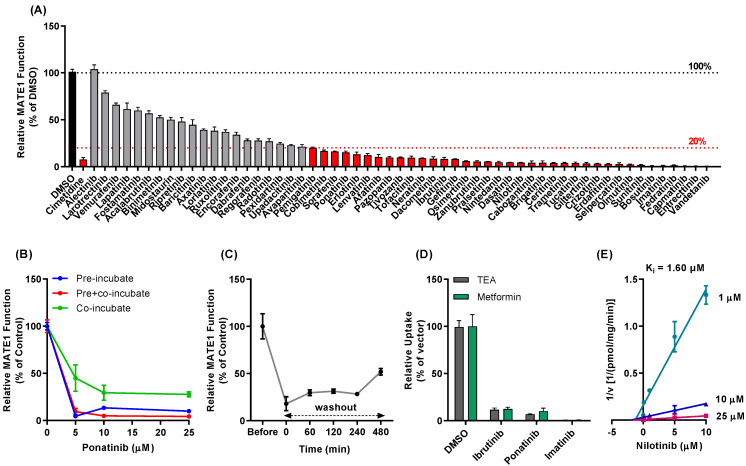
Modulation of MATE1 function by TKIs in vitro. (**A**) Different TKIs were tested at a concentration of 10 µM in both MATE1-overexpressing HEK293 cells and HEK293 cells transfected with an empty vector. The uptake of the MATE1 substrate [^14^C] TEA (2 µM) in the presence of TKIs is reported as a percent difference of uptake in cells treated with the control vehicle, DMSO. Cimetidine (25 µM) was used as a positive control inhibitor of MATE1. (**B**) HEK293 cells overexpressing MATE1 were either pre-treated with TKI (ponatinib) for 15 min before adding the substrate TEA (pre-incubate group), co-incubated with TKI and TEA (co-incubate group), or pre-treated with TKI, followed by TEA in the presence of TKI (pre + co-incubate group). (**C**) Time-dependent recovery of MATE1 function in cells pre-incubated with ibrutinib 10 µM) for 15 min, followed by removal of ibrutiniband [^14^C] TEA uptake assay was carried out at time 0, 1, 2, 4, and 8 h. Baseline MATE1 function was determined in untreated cells. (**D**) Influence of TKIs (pre + co-incubation) on comparative MATE1-dependent transport of [^14^C] TEA (2 µM) and metformin (5 µM). The graph represents relative uptake as compared to individual vector group after normalization of protein levels. Individual t-tests showed no significant difference between the two substrates. (**E**) Dixon plot of nilotinib-dependent MATE1 inhibition of varying concentrations of [^14^C] TEA. The velocity (v) was calculated based on the difference in uptake between cells with and without MATE1 overexpression. Data are shown as mean ± SEM.

**Figure 2 pharmaceutics-13-02004-f002:**
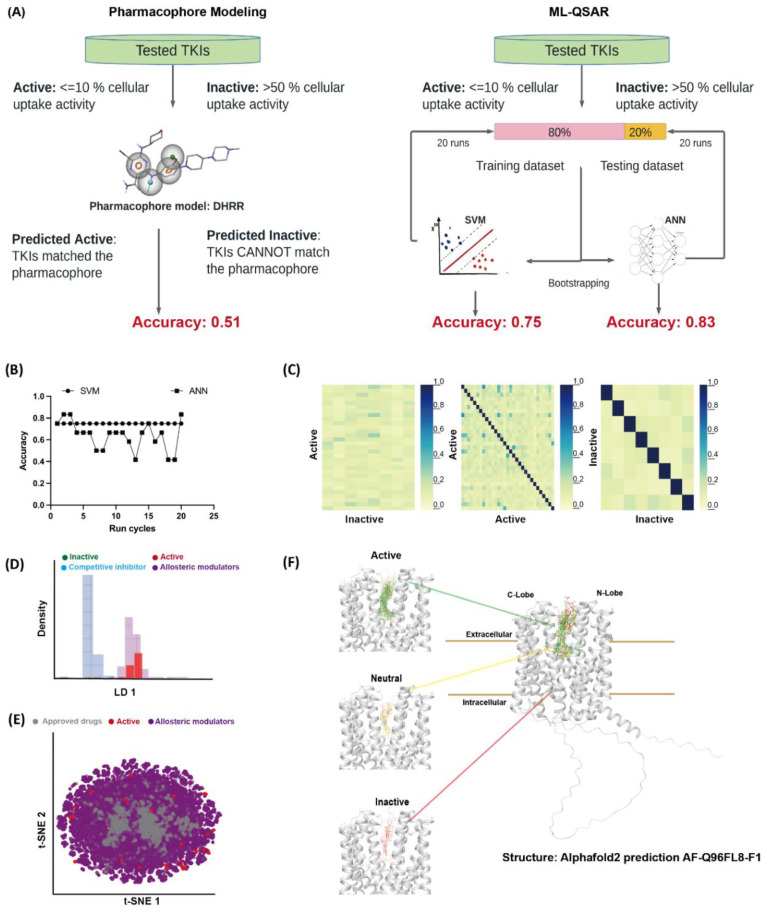
In silico analysis of mechanisms underlying MATE1 inhibition by TKIs. (**A**) The workflow of pharmacophore and ML-QSAR modeling. (**B**) The accuracy for the test dataset for 20 ML-QSAR runs. (**C**) The Tanimoto similarity comparisons of TKIs. (**D**) Probability distributions of active TKIs, inactive TKIs, allosteric modulators and competitive inhibitors using LDA. (**E**) Chemical space of the TKIs, allosteric modulators and approved drugs projected along the first two PC modes from PCA using t-SNE. (**F**) The clustering of the best binding poses for active, inactive, and neutral TKIs in a hMATE1 model. See Section 2 for details.

**Figure 3 pharmaceutics-13-02004-f003:**
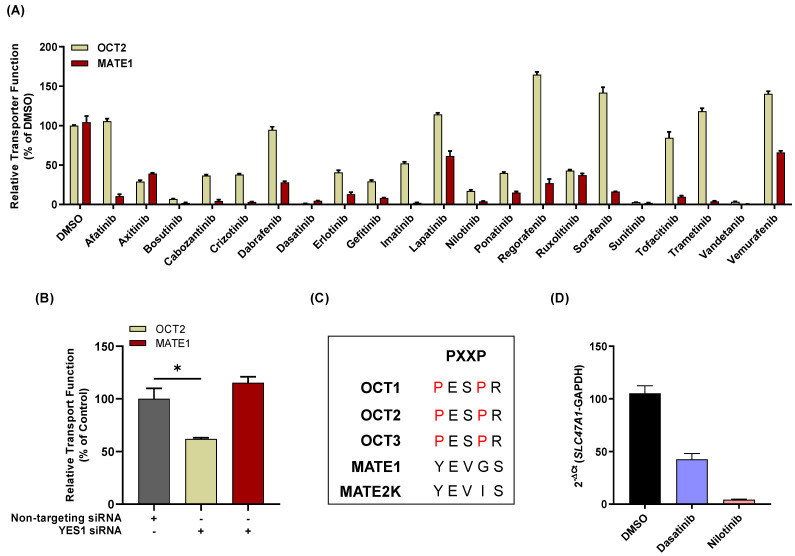
Influence of TKIs on MATE1 expression. (**A**) Relative transporter function in HEK293 cells stably transfected with human OCT2 and MATE1 [derived from Figure 1A] was evaluated by a substrate drug [^14^C] TEA in the presence of FDA approved TKIs (10 μM). The graph represents relative transport activity of indicated substrate drug compared to DMSO. (**B**) siRNA (25 nM)-mediated knockdown of YES1 kinase and its effects on HEK293 cells overexpressing human OCT2 and MATE1 using uptake assays with [^14^C] TEA. The treatment condition for each bar is mentioned below the graph. Data are shown as mean ± SEM (* *p* < 0.05). (**C**) Protein sequence of OCT1, OCT2, OCT3, MATE1 (residue 277) and MATE2K was aligned by a multiple sequence alignment program (MAFFT). (**D**) Effect of select TKIs on the expression of MATE1 gene as measured by RT-PCR (results are reported as a difference with the repective control containing empty vector).

**Figure 4 pharmaceutics-13-02004-f004:**
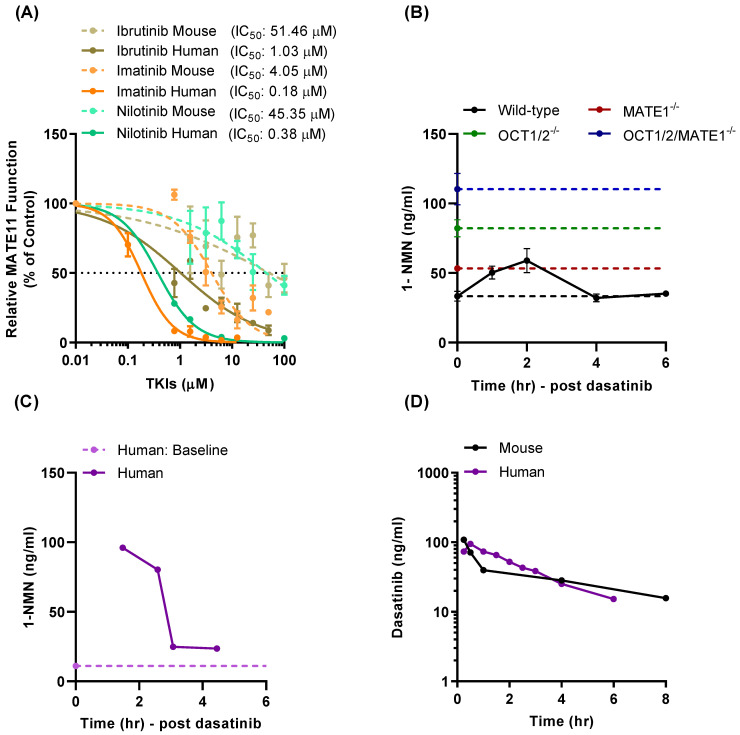
Characteristics of MATE1 inhibition by TKIs. (**A**) Comparison of IC_50_ of several TKIs in HEK293 cells overexpressing mouse or human MATE1. (**B**) Influence of mouse genotype and TKI treatment on levels of the MATE1 biomarker, N-methynicotinamide (1-NMN) in plasma. Dashed lines represent baseline levels of 1-NMN in wild-type, OCT1/2-, MATE1-, and OCT1/2/MATE1-deficient mice (n = 5 each), and solid lines represent concentration-time profiles of 1-NMN in wild-type mice after a single oral dose of dasatinib (15 mg/kg) (*p* < 0.05 treated *vs.* baseline 1-NMN level in wild-type mice). All values represent mean ± SEM. (**C**) Plasma concentration-time profile of 1-NMN in a human patient after a single oral dose (p.o.) of dasatinib (100 mg). Dashed line represents baseline 1-NMN level in plamsa. (**D**) Plasma concentration time profile of dasatinib in mice (15 mg/kg; p.o.) and a human patient (100 mg; p.o.).

**Figure 5 pharmaceutics-13-02004-f005:**
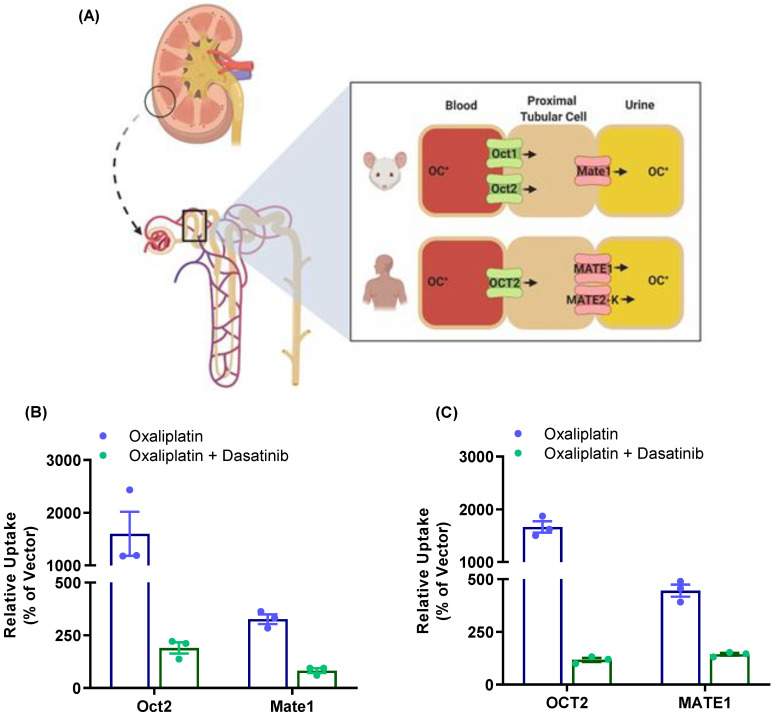
Renal transport of oxaliplatin in mice and humans. (**A**) Scheme depicting vectorial transport of organic cations (OC+) in the kidney of mice and humans. Transport activity of oxaliplatin (50 µM) was assessed in HEK293 cells overexpressing mouse (**B**) and human (**C**) OCT2, or MATE1 transporters in the presence or absence of dasatinib (10 µM).

**Figure 6 pharmaceutics-13-02004-f006:**
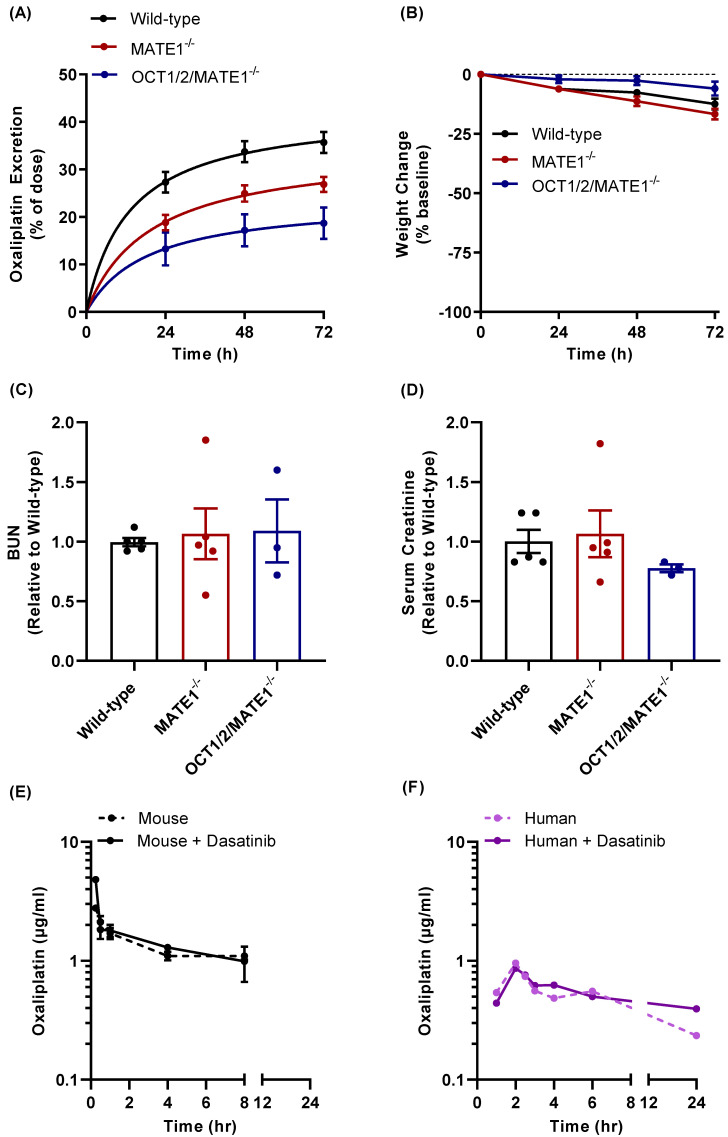
Influence of MATE1 inhibition on oxaliplatin disposition. Urinary excretion (**A**) weight loss (**B**), blood urea nitrogen (BUN) (**C**), and serum creatinine (**D**) were assessed in wild-type, MATE1-deficient, and OCT1/2/MATE1-deficient mice (n = 5 each) following a single dose of oxaliplatin (10 mg/kg, i.p.). Data represent mean ± SEM. Plasma concentration time profile of oxaliplatin in mice (10 mg/kg; i.p.) (**E**) and a human patient (85 mg/m^2^; i.v. infusion) (**F**) in the presence and absence of pre-treatment with dasatinib at the dose of 15 mg/kg in mice, and 100 mg in a human patient.

## Data Availability

All data available are reported in the article.

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
