# Peer review of "In Vitro and In Vivo Inhibition of MATE1 by Tyrosine Kinase Inhibitors"

_pharmaceutics, 2021, doi:10.3390/pharmaceutics13122004_

Round 1
Reviewer 1 Report
This is a very well written and interesting paper. The experiments are conducted appropriately, and the studies are thorough.
No major changes suggested, or additional experiments required by this reviewer.
Concerns:
My major concern is the extrapolations made to human from the in vitro or rodent data without factoring in concentrations or physiological differences. Specific comments are in the text of the document. In general -
- Would be good to add a a little more from literature on differences/similarities between Mate1 and MATE1 and substrates
- There are extrapolations made to all TKIs based on this preclinical data. The interactions with MATE1 are likely not directly associated with the pharmacological activity on tyrosine kinase but a function of the chemotype. The discussion and conclusions of each experiment should be made specifically about the chemical drug being studied and not the whole class of drugs.
- It would be good to include a justification of the concentrations used. I understand 10µM for an early screen. Were the further studies conducted at clinically relevant concentrations? Since drug-drug interactions are directly dependent upon the concentrations used, would be good to discuss the relevance of the concentrations used to what might happen in the clinic.
- The authors directly correlate the in vitro data and rodent to human – that is not appropriate without physiologic modeling.
Good Luck!
Author Response
This is a very well written and interesting paper. The experiments are conducted appropriately, and the studies are thorough.
No major changes suggested, or additional experiments required by this reviewer.
Concerns:
My major concern is the extrapolations made to human from the in vitro or rodent data without factoring in concentrations or physiological differences. Specific comments are in the text of the document. In general -
- Would be good to add a little more from literature on differences/similarities between Mate1 and MATE1 and substrates
Response 1:
Thank you for your comment. We have added a short paragraph in the introduction.
New sentence:
“The tissue distribution of OCT2 and MATE1 in mice are generally consistent with that in humans, with the exception of MATE2-K being absent in the kidney of mice (Yonezawa et. al. 2011). Human and mouse MATE1 exhibit mutual sequence identity (78%) with similar characteristics (Koepsell et. al. 2007). Substrates for MATE1 are typical organic cations e.g. TEA, 1-methyl-4-phenylpyridinium (MPP), metformin, cimetidine, procainamide, oxaliplatin etc. which are also found to be transported by mouse MATE1 (Tanihara et. al. 2007).”
a) Yonezawa A, Inui K. Importance of the multidrug and toxin extrusion MATE/SLC47A family to pharmacokinetics, pharmacodynamics/toxicodynamics and pharmacogenomics. Br J Pharmacol. 2011;164(7):1817-1825.
b) Koepsell H, Lips K, Volk C. Polyspecific organic cation transporters: structure, function, physiological roles, and biopharmaceutical implications. Pharm Res. 2007 Jul; 24(7):1227-51.
c) Tanihara Y, Masuda S, Sato T, Katsura T, Ogawa O, Inui K. Substrate specificity of MATE1 and MATE2-K, human multidrug and toxin extrusions/H(+)-organic cation antiporters. Biochem Pharmacol. 2007 Jul 15;74(2):359-71.
- There are extrapolations made to all TKIs based on this preclinical data. The interactions with MATE1 are likely not directly associated with the pharmacological activity on tyrosine kinase but a function of the chemotype. The discussion and conclusions of each experiment should be made specifically about the chemical drug being studied and not the whole class of drugs.
Response 2:
We agree with your comment. The manuscript and figure legends have been revised mentioning what specific tyrosine kinase inhibitors were used instead of highlighting whole class of drugs (TKIs).
- It would be good to include a justification of the concentrations used. I understand 10µM for an early screen. Were the further studies conducted at clinically relevant concentrations? Since drug-drug interactions are directly dependent upon the concentrations used, would be good to discuss the relevance of the concentrations used to what might happen in the clinic.
Response 3:
This is a very good point. In view of the apical localization of MATE1 in the kidney, the observed dependence of inhibitory properties of TKIs on pre-incubation conditions, and the requirement of the perpetrators to accumulate within proximal tubular cells, we believe that traditional equations used for prediction of in vivo inhibition that are derived from considerations of achievable (unbound) levels in plasma, such as those reported previously (e.g., https://accp1.onlinelibrary.wiley.com/doi/pdf/10.1002/jcph.723) are not particularly informative. As mentioned in the manuscript, the concentrations in proximal tubes are almost 10 times higher than of plasma (page 11, line 409, reference 45), also it has been reported that the intracellular concentrations of TKIs are much higher than the extracellular concentrations (administration of 25 uM imatinib results in intracellular concentrations around 4.2 mM) (Lipka et al. 2012). The notion that the administration of dasatinib to mice and the human subject is associated with measurable changes in the levels of 1-NMN provides support for the chosen strategy. We have included the cited references and the explanation to the revised manuscript (line 419).
- Lipka, D. B., Wagner, M. C., Dziadosz, M., Schnöder, T., Heidel, F., Schemionek, M., ... & Fischer, T. (2012). Intracellular retention of ABL kinase inhibitors determines commitment to apoptosis in CML cells. PloS one, 7(7), e40853.
- The authors directly correlate the in vitro data and rodent to human – that is not appropriate without physiologic modeling.
Response 4:
We agree that a physiologically based pharmacokinetics (PBPK) model would be required to provide adequate justification for extrapolation of the in vitro data to humans. We have added this correlation in the conclusion (line 515).
Reviewer 2 Report
Because tyrosine kinase inhibitors (TKI) are important for cancer treatment, drug-drug interactions of TKIs through multidrug transporters are interesting topic to date. In this study, authors investigated effects of various TKIs on MATE1 activity in vitro and in vivo. Results showed TKIs have no large effects on renal DDI related to MATE1, although considerable effects on MATE1 activity in vitro was observed.
In conclusion, results are interesting and could be published after revision.
Major
1) Dasatinib don't affect renal function as shown in figures 4-6. However, it may not true for other TKIs that is not tested in these figure. Low structural similarity between TKIs suggests mechanism to inhibit MATE1 activity is dependent on TKI. Thus, expansion of conclusion from dasatinib to other TKIs needs to be carefully. In addition, dasatinib inhibited in vitro function of MATE1. This suggests that treatment with TKIs needs caution although deleterious effect on renal function was not observed.
2) Fig. 4: Did authors examine the effects of TKI on MATE1 expression in vivo? Discrepancy between in vitro and in vivo effects may be due to gene expression. To compare results of figures 1-3, gene expression of MATE1 after TKI administration is needed. Also, reason for discrepancy between in vitro and in vivo results should be discussed.
3) Figure 1E: Number of points to distinguish competitive and non-competitive kinetics is too small. Three points are not enough. More than 4 points are required to obtain significant results. In addition, figure is too small to see kinetics.
Minors
1) Figure description, especially for figure legends throughout manuscript are not enough to understand experiments. More detailed description of figures is needed.
2) Page 2, lines 71 -: Details of expression system should be written.
3) Page 2, Lines 91 -: Assay condition is not clear especially for OCT. Composition of assay buffer should be given.
4) Page 2, Line 92: Why is assay pH for hMATE1 different from that of mMATE1? Since MATE is pH dependent transporter, results of hMATE1 and mMATE1 are difficult to compare.
5) Page 5, Lines 245-247: Figure 1B shows effect of ponatinib preincubation and no dasatinib, ibrutinib, imatinib and nilotinib.
6) Page 6, Fig.1D: Did authors preincubate cells with indicated TKIs before uptake assay as like Fig.1B?
7) Figure 1: In these experiments, authors used human MATE1 whereas mouse MATE1 was used in later figures. Please indicate animal source of MATE1 to distinguish mouse one throughout the manuscript.
8) Figure 2D: Please put labels on x and y axes.
9) Page 9, line 347: I am not sure that overexpression system is suitable for investigation of gene expression. Did authors used plasmid born expression system? If so, results shown in figure 3 doesn't indicate expression control by TKIs.
10) Fig. 3A: I suppose that MATE1 data are taken from figure 1A. If so, please describe it in the legend.
11) Fig. 3C: Please indicate residue numbers.
12) Fig. 3B and C: Figure legends may be wrong.
13) Fig. 3 title: There is no MATE1 phosphorylation data in figure 3. Title is misleading.
14) Fig. 3C: What does TEA + mean? What is control + in these experiments? Please explain them in the legend.
15) Fig. 3D: Dasatinib suppress MATE1 gene expression to ~50%. However, it completely inhibited MATE1 activity. This suggest dasatinib has multiple effect. One is gene expression, and other may be direct effect to MATE1. Please explain this.
16) When I looked figure 1B, time course of MATE1 inhibition seems to be rapid and reached maximum inhibition within 5 min. This is too fast for inhibition by gene expression. Protein expression and localization analyses is needed.
17) Fig. 4B: I think this shows NMN concentration in plasma. If so, please describe it in the legend.
18) Fig. 4B: Dashed lines for knock out mice data may be misleading. Do authors have evidence that NMN levels of KO mice are constant during time course? If there is no evidence, dashed lines must be removed.
19) Fig. 4B: Effect of MATE1 KO on NMN level is relatively small although MATE1 greatly contributes to NMN clearance. Please explain.
20) Fig. 6: Symbol color is not easy to see. I suggest you to use other color especially for blue.
Author Response
Because tyrosine kinase inhibitors (TKI) are important for cancer treatment, drug-drug interactions of TKIs through multidrug transporters are interesting topic to date. In this study, authors investigated effects of various TKIs on MATE1 activity in vitro and in vivo. Results showed TKIs have no large effects on renal DDI related to MATE1, although considerable effects on MATE1 activity in vitro was observed.
In conclusion, results are interesting and could be published after revision.
Major
1) Dasatinib don't affect renal function as shown in figures 4-6. However, it may not true for other TKIs that is not tested in these figure. Low structural similarity between TKIs suggests mechanism to inhibit MATE1 activity is dependent on TKI. Thus, expansion of conclusion from dasatinib to other TKIs needs to be carefully. In addition, dasatinib inhibited in vitro function of MATE1. This suggests that treatment with TKIs needs caution although deleterious effect on renal function was not observed.
Response 1:
We appreciate your comment. We agree that expansion of conclusion from dasatinib to other TKIs needs to be taken carefully, also further investigations are required with other TKIs in order to demonstrate the inhibition mechanism with MATE1. We have added a few sentences in line 422.
NEW Sentence:
“In view of the structural dissimilarity between TKIs, extending conclusions derived from studies with dasatinib to other TKIs is not necessarily appropriate, and suggests that further investigations are required to evaluate whether the mechanism of MTE1 inhibition is dependent on the TKI involved. In addition, while TKIs such as dasatinib potently inhibit MATE1 function in vitro, suggesting that concurrent treatment with such TKIs warrants caution, these interactions are not necessarily associated with a deleterious effect on renal function.”
2) Fig. 4: Did authors examine the effects of TKI on MATE1 expression in vivo? Discrepancy between in vitro and in vivo effects may be due to gene expression. To compare results of figures 1-3, gene expression of MATE1 after TKI administration is needed. Also, reason for discrepancy between in vitro and in vivo results should be discussed.
Response 2:
Thank you for your thoughtful comment. We have not pursued these experiments yet, however, we agree that this aspect requires further attention to an extent that is beyond the scope of this manuscript. In particular, such studies would involve investigation of MATE1 protein expression changes in vivo along with mRNA, as well as effects on protein localization after exposure to TKIs given at an optimized dose, schedule, and route of administration, with careful attention to time points of data collection. These studies are currently under investigation.
3) Figure 1E: Number of points to distinguish competitive and non-competitive kinetics is too small. Three points are not enough. More than 4 points are required to obtain significant results. In addition, figure is too small to see kinetics.
Response 3:
Thank you for your comment. We have modeled our study design based on recently published work on the interaction of TKIs with other transporters in highly regarded journals (eg, PMID: 33664059; PMID: 26979622; PMID: 33495337), and would like to point out the existence of a recent precedent in this journal (eg, https://www.ncbi.nlm.nih.gov/pmc/articles/PMC8470274/), employing a strategy very similar to ours. Based on these considerations, we propose to retain the information as originally presented, while acknowledging that more accurate information could have been obtained with the use of more stringent study designs. In addition, we have revised the figure and added an extra concentration of an inhibitor, nilotinib, at 5 µM to demonstrate the non-competitive inhibition mechanism.
Minors
1) Figure description, especially for figure legends throughout manuscript are not enough to understand experiments. More detailed description of figures is needed.
Response 1:
Thank you so much for your comment. We have provided a more detailed description of the figures throughout the manuscript.
2) Page 2, lines 71 -: Details of expression system should be written.
Response 2:
We appreciate your comment regarding explaining the expression system. We have added the details in section 2.1 Cellular models and cell culture conditions (line 68)
NEW sentence:
“The cDNAs for the mouse and human plasmids of OCT2 or MATE1 were obtained from Origene (Rockville, MD), and the reconstructed cDNAs were subcloned into an empty vector containing pcDNA5/FRT. The vector was transfected into HEK293 cells using the Flp-In system (Invitrogen) and selected for expression using geneticin (G418).”
3) Page 2, Lines 91 -: Assay condition is not clear especially for OCT. Composition of assay buffer should be given.
Response 3:
Thank you for your comment. Assay condition for MATE1 and composition of transport buffer has been mentioned in line 101.
OLD sentence:
For MATE1-overexpressed cells, cells were incubated in medium containing ammonium chloride (pH 7.4 for hMATE1 and pH 8.4 for mMATE1) to ensure interactions can be evaluated between outward-facing MATE1 and substrate uptake before adding inhibitors.
NEW sentence:
“For MATE1-overexpressed cells, cells were incubated in medium containing 30 mM ammonium chloride for 20 minutes (pH 7.4 for hMATE1 and pH 8.4 for mMATE1) to ensure interactions can be evaluated between outward-facing MATE1 and substrate uptake before adding inhibitors. The composition of transport buffer was as follows: 145 mM NaCl, 3 mM KCl, 1 mM CaCl2, 0.5 mM MgCl2, 5 mM D-Glucose, and 5 mM HEPES. Although the physiological function of MATE1 is to support luminal efflux, previous studies have shown that kinetic parameters of MATE1-mediated transport are not significantly different between the outward (uptake) and inward (efflux) orientations [14].”
4) Page 2, Line 101: Why is assay pH for hMATE1 different from that of mMATE1? Since MATE is pH dependent transporter, results of hMATE1 and mMATE1 are difficult to compare.
Response 4:
Thank you for your thoughtful comment. We followed previously published uptake protocols of mMATE1 by Kobara et al 2008 (a) and Hiasa et al 2006 (b). While we acknowledge that minor changes to the experimental conditions might have influenced the results in the absence of inhibitors, it seems unlikely that this impacted the degree to which TKIs can differentially influence the function of mMATE1 and hMATE1. Most importantly, the notion that mMATE1 is less sensitive to inhibition by TKIs than hMATE1 is consistent with prior published data, as mentioned in our manuscript.
a) Kobara A, Hiasa M, Matsumoto T, Otsuka M, Omote H, Moriyama Y. A novel variant of mouse MATE-1 H+/organic cation antiporter with a long hydrophobic tail. Arch Biochem Biophys. 2008;469(2):195-199.
b) Hiasa M, Matsumoto T, Komatsu T, Moriyama Y. Wide variety of locations for rodent MATE1, a transporter protein that mediates the final excretion step for toxic organic cations. Am J Physiol Cell Physiol. 2006;291(4):C678-C686.
5) Page 5, Lines 245-247: Figure 1B shows effect of ponatinib preincubation and no dasatinib, ibrutinib, imatinib and nilotinib.
Response 5:
Thank you for your comment. We have revised the manuscript to more clearly indicate which studies were performed with which TKIs.
6) Page 6, Fig.1D: Did authors preincubate cells with indicated TKIs before uptake assay as like Fig.1B?
Response 6:
Yes, cells were pre-incubated with TKIs for 15 min followed by the substrate uptake of TEA and metformin.
New sentence in line 279:
“Influence of TKIs (pre+co-incubation) on comparative MATE1-dependent transport of TEA (2 µM) and metformin (5 µM).”
7) Figure 1: In these experiments, authors used human MATE1 whereas mouse MATE1 was used in later figures. Please indicate animal source of MATE1 to distinguish mouse one throughout the manuscript.
Response 7:
Thank you for your comment. Revision has been made, and we have used “human MATE1” and “mouse MATE1” to differentiate the two groups in the manuscript.
8) Figure 2D: Please put labels on x and y axes.
Response 8:
Label in Figure 2D has been added.
9) Page 9, line 347: I am not sure that overexpression system is suitable for investigation of gene expression. Did authors used plasmid born expression system? If so, results shown in figure 3 doesn't indicate expression control by TKIs.
Response 9:
Thank you for your comment. To clarify, the data presented here is the difference of expression changes between MATE1 overexpressed cells and the cells transfected with vector control alone. We have added a description of that in the figure legends. Further investigations are ongoing and analyzing MATE1 gene expression with TKIs would be ideal in other experimental models such as renal proximal tubular cells. A sentence has been added in line 365.
10) Fig. 3A: I suppose that MATE1 data are taken from figure 1A. If so, please describe it in the legend.
Response 10:
Yes, MATE1 data has been taken from Figure 1 (A) which has been mentioned in the figure legend.
NEW Sentence:
“Comparative inhibition of OCT2 and MATE1 [derived from Figure 1 (A)] by different TKIs (10 µM).”
11) Fig. 3C: Please indicate residue numbers.
Response 11:
The corresponding proline-rich region (PXXPR) is located at 277 residue for MATE1, which has been added to the figure legend.
New Sentence:
“Protein sequence of OCT1, OCT2, OCT3, MATE1 (residue 277), and MATE2K was aligned by a multiple sequence alignment program (MAFFT).”
12) Fig. 3B and C: Figure legends may be wrong.
Response 12:
Thank you for your comment. Figure legends have been changed.
13) Fig. 3 title: There is no MATE1 phosphorylation data in figure 3. Title is misleading.
Response 13:
We apologize for this mistake. Figure 3 Title has been corrected.
New Sentence:
Figure 3. Influence of TKIs on MATE1 expression.
14) Fig. 3C: What does TEA + mean? What is control + in these experiments? Please explain them in the legend.
Response 14:
TEA + refers to the condition in the presence of TEA. Since TEA was present in all groups, we removed the + symbols, and have modified the figure legend accordingly.
15) Fig. 3D: Dasatinib suppress MATE1 gene expression to ~50%. However, it completely inhibited MATE1 activity. This suggest dasatinib has multiple effect. One is gene expression, and other may be direct effect to MATE1. Please explain this.
Response 15:
We agree that dasatinib might have multiple effects and one possible reason could be due to its different potency against MATE1 expression.
NEW Sentence in line 361:
“Although dasatinib completely inhibited the MATE1-mediated uptake of TEA, the suppression of gene expression observed in the presence of dasatinib was ~50%, suggesting that inhibition might occur via different mechanisms.”
16) When I looked figure 1B, time course of MATE1 inhibition seems to be rapid and reached maximum inhibition within 5 min. This is too fast for inhibition by gene expression. Protein expression and localization analyses is needed.
Response 16:
We agree with your comment that additional experimentation, beyond the scope of this present manuscript, is required to further elucidate the mechanism. This would include examination of protein expression and localization changes following exposure to dasatinib in different models in vitro and in vivo. These studies are presently ongoing.
17) Fig. 4B: I think this shows NMN concentration in plasma. If so, please describe it in the legend.
Response 17:
Yes, the 1-NMN concentration shown in different mouse models was in plasma which has been added in the figure legend.
NEW sentence:
“Influence of mouse genotype and TKI treatment on levels of the MATE1 biomarker, N-methynicotinamide (1-NMN) in plasma.”
18) Fig. 4B: Dashed lines for knock out mice data may be misleading. Do authors have evidence that NMN levels of KO mice are constant during time course? If there is no evidence, dashed lines must be removed.
Response 18:
We have not experimentally verified that NMN levels remain constant over the time course of sample collection in KO mice, for example following the administration of a mock (vehicle control). In order to make the figure more easily interpretable, we propose to retain the dashed line as a representation of the baseline level of NMN in knockout mice, while the solid line represents the concentration of NMN in plasma of wild-type mice after receiving dasatinib. Clarification has been made in the figure legend to avoid confusion from readers.
OLD sentence:
Data represent baseline levels in OCT1/2-, MATE1-, and OCT1/2/MATE1-deficient mice (n=5 each) and concentration-time profiles in wild-type mice after a single oral dose of dasatinib (15 mg/kg).
NEW sentence:
“Dashed lines represent baseline levels of 1-NMN in wild-type, OCT1/2-, MATE1-, and OCT1/2/MATE1-deficient mice (n=5 each) and solid lines represent concentration-time profiles of 1-NMN in wild-type mice after a single oral dose of dasatinib (15 mg/kg).”
19) Fig. 4B: Effect of MATE1 KO on NMN level is relatively small although MATE1 greatly contributes to NMN clearance. Please explain.
Response 19:
In our result, we have mentioned that 1-NMN is a dual endogenous substrate of both OCT2 and MATE1. Even though MATE1 greatly contributes to 1-NMN clearance, the level of 1-NMN in MATE1 KO mice is relatively small compared to OCT1,2 and OCT1,2/MATE1-deficient mice which could be due to the accumulation in kidney. It would be worth investigating to measure the concentration of 1-NMN in MATE1-deficient mouse kidney.
NEW Sentence in line 407:
“Even though MATE1 greatly contributes to 1-NMN clearance, the level of 1-NMN in MATE1-deficient mice is relatively small compared to OCT1,2 and OCT1,2/MATE1-deficient mice which could be due to its accumulation in the kidney. Further investigation is required to measure the concentration of 1-NMN in MATE1-deficient mouse kidney.”
- Miyake T, Kimoto E, Luo L, et al. Identification of Appropriate Endogenous Biomarker for Risk Assessment of Multidrug and Toxin Extrusion Protein-Mediated Drug-Drug Interactions in Healthy Volunteers. Clin Pharmacol Ther. 2021;109(2):507-516.
20) Fig. 6: Symbol color is not easy to see. I suggest you to use other color especially for blue.
Response 20:
Thank you so much for your comment. The color has been changed in Figure 6.
Reviewer 3 Report
This is an interesting article where the authors examined the interaction of FDA approved tyrosine kinase inhibitors (TKIs) in the renal transporter MATE1 (SLC47A1) and measure their influences on the elimination and toxicity of the MATE1 substrate oxaliplatin using in vitro in cells, in silico by molecular docking simulations, and in vivo in genetically-engineered mouse models and a patient with cancer. The manuscript is well structured and descriptive. The finding is significant and the manuscript could be accepted for publication after minor corrections of English grammar and sentence redactions.
- Author should provide full abbreviation of TEA in lane 273.
- Did author check the time dependent recovery of MATE1 function more than 8 hrs as shown in fig 1C?
Author Response
This is an interesting article where the authors examined the interaction of FDA approved tyrosine kinase inhibitors (TKIs) in the renal transporter MATE1 (SLC47A1) and measure their influences on the elimination and toxicity of the MATE1 substrate oxaliplatin using in vitro in cells, in silico by molecular docking simulations, and in vivo in genetically-engineered mouse models and a patient with cancer. The manuscript is well structured and descriptive. The finding is significant and the manuscript could be accepted for publication after minor corrections of English grammar and sentence redactions.
1. Author should provide full abbreviation of TEA in lane 273.
Response 1:
Thank you for your comment. We have added the full abbreviation TEA in line 81.
2. Did author check the time dependent recovery of MATE1 function more than 8 hrs as shown in fig 1C?
Response 2:
Thanks for your comment. We have not evaluated time points beyond 8 hours for MATE1 recovery since the experiment was designed to demonstrate the presence or absence of rapid reversibility (i.e., within less than 8 hours). Ongoing studies will focus on a more detailed characterization of this profile for a larger group of TKIs.
Round 2
Reviewer 2 Report
Most of my questions are adequately addressed. However, there is a problem that must be solved before publishing.
1) Transport assay condition for MATE1
I am confused since assay condition for MATE1 containing 30 mM of ammonium chloride. It increased cytoplasmic pH and completely inhibits MATE1 activity as shown in Ref 18. In addition, both of Ref 17 and 18 used pH 8.0 for TEA uptake. Assay condition would be incorrect. Please recheck condition. This is important point for readers to avoid misunderstanding.
2) Line 426: MTE1 should be MATE1
Author Response
Most of my questions are adequately addressed. However, there is a problem that must be solved before publishing.
1) Transport assay condition for MATE1
I am confused since assay condition for MATE1 containing 30 mM of ammonium chloride. It increased cytoplasmic pH and completely inhibits MATE1 activity as shown in Ref 18. In addition, both of Ref 17 and 18 used pH 8.0 for TEA uptake. Assay condition would be incorrect. Please recheck condition. This is important point for readers to avoid misunderstanding.
Response 1:
Thank you for your comment and we appreciate your concern. In order to address the issue, we have revisited our protocol and conducted separate uptake studies with TEA at different pH (7.4 and 8.4) in HEK293 cells overexpressing human (Fig A) and mouse (Fig B) MATE1 [please see the attachment]. Moreover, Ref 18 does not contradict our conclusion since other manuscripts (reference below) have successfully used a similar condition to study MATE1 function while pre-incubating cells with 30mM NH4Cl for 20 min and conducting uptake experiment of substrate compounds with transport media (without NH4Cl). Thus, co-incubation with NH4Cl was not included in our study design.
In addition, Terada et al. 2006 (Fig 2B) demonstrated that intracellular acidification induced by 30 mM NH4Cl resulted in a marked stimulation of TEA uptake. Also, Chu et al 2016 reported that a shift of IC50 was observed in MATE1 cells pre-treated with (0.8 µM) and without (15.4 µM) NH4Cl.
Figure [please see the attachment]: Uptake of TEA in HEK293 cells overexpressing human (A) and mouse (B) MATE1. Cells were pre-incubated with transport media containing 30mM NH4Cl for 20 min. Then, pre-incubation media was removed, and cells were co-incubated with transport media containing radiolabeled TEA with or without 30 mM NH4Cl for 15 min. The graph represents the relative transport activity of TEA compared to vector control. Data are shown as mean ± SEM.
- Terada, T., Masuda, S., Asaka, J., Tsuda, M., Katsura, T., & Inui, K. (2006). Molecular cloning, functional characterization and tissue distribution of rat H+/organic cation antiporter MATE1. Pharmaceutical research, 23(8), 1696–1701.
- Nakamura, T., Yonezawa, A., Hashimoto, S., Katsura, T., & Inui, K. (2010). Disruption of multidrug and toxin extrusion MATE1 potentiates cisplatin-induced nephrotoxicity. Biochemical pharmacology, 80(11), 1762–1767.
- Chu, X., Bleasby, K., Chan, G. H., Nunes, I., & Evers, R. (2016). The Complexities of Interpreting Reversible Elevated Serum Creatinine Levels in Drug Development: Does a Correlation with Inhibition of Renal Transporters Exist?. Drug metabolism and disposition: the biological fate of chemicals, 44(9), 1498–1509.
- Guo, D., Yang, H., Li, Q., Bae, H. J., Obianom, O., Zeng, S., Su, T., Polli, J. E., & Shu, Y. (2018). Selective Inhibition on Organic Cation Transporters by Carvedilol Protects Mice from Cisplatin-Induced Nephrotoxicity. Pharmaceutical research, 35(11), 204.
We also agree that Ref 17 (currently Ref 20) and 18 did the TEA uptake at pH 8.0, however, both studies Ref 17 (Fig 2D) and 18 (Fig 3C) demonstrated the pH-dependent uptake of TEA in HEK293 cells overexpressing mMATE1 which have been observed to be increased at higher pH (pH range 8.0 - 8.5). Therefore, we carried our uptake experiments out for mMATE1 at pH 8.4. In contrast, several uptake studies were conducted in cells overexpressing hMATE1 using a transport medium containing pH 7.4 [Kajiwara et al 2009 (Fig 2); Omote et al 2018; George et al 2021].
Therefore, in order to avoid readers' confusion, we have revised sentences in the method section in line 100, added references for hMATE1, and replaced Ref 18 (currently Ref 21).
- Kajiwara, M., Terada, T., Ogasawara, K., Iwano, J., Katsura, T., Fukatsu, A., Doi, T., & Inui, K. (2009). Identification of multidrug and toxin extrusion (MATE1 and MATE2-K) variants with complete loss of transport activity. Journal of human genetics, 54(1), 40–46.
- Omote, S., Matsuoka, N., Arakawa, H., Nakanishi, T., & Tamai, I. (2018). Effect of tyrosine kinase inhibitors on renal handling of creatinine by MATE1. Scientific reports, 8(1), 9237.
- George, B., Wen, X., Jaimes, E. A., Joy, M. S., & Aleksunes, L. M. (2021). In Vitro Inhibition of Renal OCT2 and MATE1 Secretion by Antiemetic Drugs. International journal of molecular sciences, 22(12), 6439.
Old Sentence:
For MATE1-overexpressed cells, cells were incubated in medium containing 30 mM ammonium chloride (pH 7.4 for hMATE1 and pH 8.4 for mMATE1) following previously published protocol to ensure interactions can be evaluated between outward-facing MATE1 and substrate uptake before adding inhibitors [17], [18]. The composition of transport buffer was as follows: 145 mM NaCl, 3 mM KCl, 1 mM CaCl2, 0.5 mM MgCl2, 5 mM D-Glucose, and 5 mM HEPES.
NEW Sentence:
“MATE1-overexpressed cells were pre-incubated with transport media containing 30 mM ammonium chloride for 20 min at 37°C following previously published protocol (pH 7.4 for hMATE1 [17] - [19] and pH 8.4 for mMATE1 [20], [21]) to ensure interactions can be evaluated between outward-facing MATE1 and substrate uptake before adding inhibitors. Then, preincubation media was removed and cells were incubated with transport media (NH4Cl free) containing radiolabeled compounds. The composition of transport media was as follows: 145 mM NaCl, 3 mM KCl, 1 mM CaCl2, 0.5 mM MgCl2, 5 mM D-Glucose, and 5 mM HEPES.”
2) Line 426: MTE1 should be MATE1
Response 2:
Thank you for your comment. The correction has been made in line 426.
